# Do exhausted primary school students cheat more? A randomized field experiment

**Tamás Keller**[ID][1,2,3]*, **Hubert János Kiss**[ID][2,4]

1 Computational Social Science–Research Center for Educational and Network Studies, Centre for Social Sciences, Budapest, Hungary, 2 Institute of Economics, Centre for Economic and Regional Studies, Budapest, Hungary, 3 TÁRKI Social Research Institute, Budapest, Hungary, 4 Corvinus University of Budapest, Budapest, Hungary

* keller.tamas@tk.hu

**Data Availability Statement:** The data are available at the OSF platform: https://osf.io/2ykp8/wiki/home/.

**Funding:** T.K. and H.J.K. acknowledge support from the Institute of Economics' internal grant that

## Abstract

Motivated by the two-decade-long scientific debate over the existence of the ego-depletion effect, our paper contributes to exploring the scope conditions of ego-depletion theory. Specifically, in a randomized experiment, we depleted students' self-control with a cognitively demanding task that required students' effort. We measured the effect of depleted self-control on a subsequent task that required self-control to not engage in fraudulent cheating behavior—measured with an incentivized dice-roll task—and tested ego-depletion in a large-scale preregistered field experiment that was similar to real-life situations. We hypothesized that treated students would cheat more. The data confirms the hypothesis and provides causal evidence of the ego-depletion effect. Our results provide new insights into the scope conditions of ego-depletion theory, contribute methodological information for future research, and offer practical guidance for educational policy.

## 1. Introduction

The idea that people can consume their stock of self-control in situations that require self-control, resulting in a reduction in self-control in a subsequent self-control task, is referred to as ego-depletion. The ego-depletion theory is an elegant theory in social psychology that has led to a blossoming of empirical research in this field [1]. However, after more than two decades of research, there is still no conclusive agreement between skeptics and proponents of the theory that the ego-depletion effect is real [2].

Early meta-analyses on ego depletion reveal medium-to-large effect sizes that suggest that exhausted self-control leads to impaired performance in subsequent self-control tasks [3]. Later, the theory faced serious threats, as subsequent meta-analysis showed that the effect does not differ empirically from zero [4]. The finding of a large multi-lab registered replication report confirmed the nill ego-depletion effect in an inference task requiring cognitive control and attention [5]. More recently, however, Friese et al. [2] summarized that while the criticisms seriously challenge the ego-depletion theory, these criticisms are not sufficient to conclusively debunk it. Based on these considerations, some features of prior empirical research require revision, so as to explore the specific conditions under which ego-depletion effect might occur.

supported innovative research ideas during the COVID-19 epidemic. T.K. acknowledges support from the Hungarian National Research, Development and Innovation Office NKFIH (grant number K-135766), from the János Bolyai Research Scholarship of the Hungarian Academy of Sciences (BO/00569/21/9) and from the ÚNKP-21-5 New National Excellence Program of the Ministry for Innovation and Technology from the source of the National Research, Development and Innovation Fund. H.J.K. acknowledges support from the Hungarian National Research, Development and Innovation Office NKFIH (grant number K-119683).

**Competing interests:** The authors have declared that no competing interests exist

First, some previous research might have employed weak experimental manipulation due to the adopted sequential task paradigm, which deploys two consecutive tasks. While both tasks require self-control in the treatment group, only the second task requires it in the control group. Within the sequential task paradigm, the depletion of self-control hinges on the assumption that participants exert sufficient effort to deplete their self-control resources. Nevertheless, experimental manipulations are often abstract and tiresome cognitive tasks. They may either prompt respondents to control automatic reactions beyond individual awareness (suppress thoughts) or require effort to carry out some tedious exercises. Treated people might not be sufficiently motivated to engage with the manipulation task, which can cause weak experimental manipulation [6]. Therefore, new research should explore the ego-depletion effect by employing experimental manipulation beyond the sequential task paradigm.

Second, previous research employed various outcome variables; each of these might incorporate additional aspects besides self-control. The reliability of the outcome variables concerning self-control is often not explained, and the possible interplay between the additional aspects and self-control is usually not discussed. This shortcoming requires the usage of (new) outcome variables with an intuitive direct link to self-control.

Third, most prior research has tested ego depletion in lab experiments that provide artificial circumstances that are sometimes quite different from real-life situations. By contrast, various real-life situations offer tangible examples of how depletion triggers loss of self-control. Therefore, there is a need to move from lab to field experiments when testing the ego-depletion effect, thereby testing depletion in respondents' real-life contexts.

To address these issues, we conducted a large-scale online experiment including 1,143 primary school students from 126 classrooms in rural Hungarian primary schools. Our paper contributes to exploring the scope conditions of ego-depletion theory in the following ways.

First, we go beyond the sequential task paradigm. We depleted treated students' self-control using an approximately 20-minute questionnaire. This contained a 10-minute grade-specific math test and subsequent questions challenging students' ability to delay gratification and altruism. Our experimental manipulation is based on the suggestion by Baumeister and Vohs [7] that solving complex logic problems consumes energy and leads to impaired self-control. In particular, math problems have been used in prior studies to deplete self-control [4]. After the 20-minute-long questionnaire (depletion task), students in the treatment group received the second task (outcome measure) requiring self-control. Students in the control group received only the second task, and they did not solve the 20-minute-long questionnaire beforehand.

Second, our outcome measure is cheating to achieve a more valuable outcome. In our experimental situation, students could gain a more valuable gift if they behaved dishonestly, that is, if they did not exert self-control and chose a more appealing gift available to them by not telling the truth. Cheating is a policy-relevant outcome that might be of interest to educational practitioners. Specifically, depleted students might cheat in school—to attain a desired grade, for instance [8]. Yet, little research has been conducted on cheating, mainly in the context of university rather than compulsory education [9, 10]. Prior experimental economic literature involving primary school students has almost exclusively explored the associations between socioeconomic factors and cheating [11, 12], with little attention to the social-psychological foundation of cheating. In sum, cheating as an outcome variable offers the opportunity to test students' ability to exert self-control. In particular, it generates new substantive results in primary school students that might be interesting to educational policymakers.

Third, we test our hypothesis with a field experiment that explores ego depletion in a real rather than artificial situation, as the exam situation (solving math problems) is familiar to all students. Moreover, we deployed a large-scale and preregistered study. These features are

important, since prior empirical results of ego-depletion were achieved with small sample sizes, and therefore, skeptics of the ego-depletion theory condemn these results as driven by specification-search (p-hacking) and small-study effects [13].

Our findings support the prediction of the ego-depletion effect. The exhausted (that is, treated) students, whose stock of self-control had been depleted by solving the 20-minute-long questionnaire, cheated by approximately 4.4 percentage points more than the control students. Our results suggest that our manipulation, which resembled the depletion that students might face in their everyday school context, consumes students' self-control and leaves them unable to resist the temptation to behave dishonestly.

Our results contribute to determining the scope conditions of the ego-depletion effect. Ego depletion occurs when primary school students' self-control is depleted by a cognitively demanding task (1), when dishonest behavior—i.e., cheating—is examined (2), and when ego-depletion is observed in the real-life context instead of in artificial laboratory circumstances (3). The practical consequence emerging from our work is that if primary school students are depleted during cognitively demanding exercises, they cheat more—an important finding that practitioners of education should consider when they plan students' daily school schedules.

The results imply that dishonesty, not honesty, is intuitive, as resisting the temptation to be dishonest requires resources that might be consumed in tasks requiring self-control. Nevertheless, our paper has a narrow focus on the ego-depletion effect. We acknowledge, but do not directly focus on, the more general scientific debates on intuitive honesty/dishonesty [14] and the related tests on time pressure [15, 16] or cognitive load [17] that bring (sometimes conflicting) experimental evidence to this debate.

## 2. Materials and methods

### 2.1 Research transparency

We followed our detailed preanalysis plan, which we archived at the Open Science Forum (OSF) before receiving the endline data (https://osf.io/jhms2/). The data and analytic scripts to reproduce the analyses are available on the OSF page of the project: https://osf.io/2ykp8/.

### 2.2 Study overview

We conducted a large-scale online experiment among Hungarian primary school students between May 18 and June 8 2020. Primary education in Hungary is compulsory and encompasses the primary and lower-secondary ISCED 1 and ISCED 2 levels, comparable to elementary and middle school in the United States.

We recruited our respondents from an ongoing research program with 2,898 students (grades 4–8) in 29 Hungarian primary schools. We contacted and asked students via their schools to participate in a voluntary online experiment.

We reached a self-selected sample of students. Students participated in the experiment at home (students were undertaking online home-based education in response to the Covid-19 pandemic) under unsupervised conditions. On average, students who participated in our survey had better grades and better school behavior than their classmates—features typical of motivated students (see S1 Table).

The self-selected students who participated in our survey may have differed from their non-respondent classmates in motivation—that is, in terms of the motivation that may have prompted students to engage with the purpose of the study and self-select into the sample. Motivation might render immunity to ego depletion, but only if the experienced depletion is mild—suggesting that there is a limit to the influence of motivation on ego-depletion [18].

**Table 1. Descriptive statistics.**

| | Baseline variables measured before the treatment | | | | | | | | Treatment | Outcome variables: Cheating | | Less valuable gift [fatigue] [g] |
| --- | --- | --- | --- | --- | --- | --- | --- | --- | --- | --- | --- | --- |
| | Girl | Age[a] | N of books[b] | GPA[c] | Disruptive school beh.[d] | Math test | DG[e] | Altruism[f] | Depleted students | All misreports | More valuable gift[g] | |
| Mean | 0.499 | 12.82 | 0 | 3.737 | 1.331 | 0.676 | 0.814 | 0.852 | 0.690 | 0.126 | 0.089 | 0.045 |
| SD | 0.500 | 1.432 | 1 | 0.983 | 0.382 | 0.266 | 0.389 | 0.355 | 0.463 | 0.332 | 0.284 | 0.207 |
| Min | 0 | 9.781 | -0.724 | 1 | 1 | 0 | 0 | 0 | 0 | 0 | 0 | 0 |
| Max | 1 | 16.32 | 3.226 | 5 | 3.375 | 1 | 1 | 1 | 1 | 1 | 1 | 1 |
| N | 1143 | 1143 | 1106 | 1127 | 1025 | 983 | 983 | 983 | 1143 | 1143 | 1096 | 1046 |
| Missing % | 0 | 0 | 3.24 | 1.4 | 10.32 | 14 | 14 | 14 | 0 | 0 | 0.0885 | 0.045 |

[a] Students' age refers to their age at the experiment.

[b] Assessed by the following question in the students' questionnaire: "How many books do you have? You should count the number of books you and your parents possess together. Please do not include your coursebooks and newspapers" (answer categories: less than one shelf 0–50; one shelf ca. 50; 2–3 shelves (ca: 150); 4–6 shelves (ca: 300); 2 bookcases (ca: 300–600 books); 3 bookcases (ca: 600–1000 books); more than 1000 books. The variable was z-standardized to 0 mean and 1 standard deviation.

[c] Grades are integers between 1 (worst) and 5 (best). Grades reported in this table are teacher-awarded grades reported by the teachers. The source of grades is students' mid-term reporting cards issued in January 2020.

[d] An index was calculated from the mean of the following eight disruptive school behaviors: teasing others, playing or reading something, being noisy, walking around, eating or chewing gum, sending letters, talking or laughing, being late. We asked about the frequency of these behaviors by using the following scale: 1 = "Never"; 2 = "Sometimes"; 3 = "Frequently"; 4 = "Almost always." We measured this variable in a baseline teacher survey in February 2020, when homeroom teachers answered this question for all students in their class.

[e] DG refers to the delay of gratification. It was measured in a not incentivized, real choice situation where students chose between a more valuable future outcome and a less valuable immediate outcome. Students saw a picture of colorful wristbands and were asked the following question: "Do you want to have one wristband now, or two wristbands tomorrow?" Immediate gratification was coded as 0; delayed gratification was coded as 1. We measured this variable in a baseline online survey in April 2020.

[f] We measured altruism with the following questions: "Imagine that you are going to the zoo with some of your classmates. One of your classmates has forgotten to bring money for the entrance ticket. You have enough money for two entrance tickets. Would you lend your classmate the money for the entrance ticket?" Altruism is binary variable = 1 if the student lent money and 0 otherwise. The category "I do not know" was coded as zero. We measured this variable in a baseline online survey in April 2020.

[g] Missingness is due to the preregistered decision rule that restricted our analysis in the cases where students opted for a more/less valuable gift on their individualized preference list than they rolled.

Therefore, our reliance on a sample containing self-selected motivated primary school students might have introduced limited bias in estimating the ego-depletion effect.

The analytical sample contains 1,143 students in 126 4th through 8th grade classrooms at 28 schools. Descriptive statistics about the main variables in the paper are presented in Table 1. Half of the students (49.9%) were female (N = 570). Students' age ranged from 10 to 16 (mean = 12.82, standard deviation = 1.43). The high maximum age reflects the students who had to repeat classes.

The participating students were not representative of the Hungarian student population in terms of students' academic achievement and social background. On average, students' test scores and socioeconomic status are lower in our sample than the Hungarian average (see S2 Table). Corresponding with the participating students' relatively disadvantaged social background, half of them (51%, N = 582) are from small rural settlements with fewer than 3,000 inhabitants.

The demographic composition of the analytical sample leads to an over-representation of students with weaker academic achievement and from poorer social backgrounds. Thus, students in our sample might have lower initial self-control, since low academic achievement

translates into low self-control [19]. Nevertheless, the ego-depletion effect is weaker among low self-control students who have experience in resisting acute temptation and have gained experience in how to resist temptation (in contrast to high self-control students who might lack these experiences) [20]. Therefore, our estimates of the ego-depletion effect in our sample are conservative.

## 2.3 The measurement and coding of the outcome variable: Cheating

We measured our outcome (cheating) with a dice-roll task (a modified version of the standard dice-roll experiment [21]) to collect individual data about students' dishonest behavior. Our procedure consisted of the following steps:

1. We asked students to create their preference list by ranking six different objects of different monetary values according to the *subjective* value they attached to each object. We told the students that the rank of objects on the preference list should correspond to their desire for the object. For example, the first object on the preference list should be the object they most desire and would most like to receive as a gift. The six objects were a pencil case, a pouch, a mug, a pen, a keyring, and a badge, as shown in Fig 1.

2. We asked students to roll a virtual six-sided dice and instantly reveal the rolled number.

3. We showed students their subjective preference list and informed them how each object on their list would correspond to specific numbers between 1 and 6. For example, students were informed that 1 corresponded to their most desired object; 2 corresponded to the second object on the preference list, and 6 corresponded to the least desired object.

4. We asked students to report the *rolled* number. We indicated that they would receive the object based on the *reported* number. Therefore, once we asked for the rolled number, students could report any number between 1 and 6 and the number students reported could differ from the rolled number.

In sum, we tempted students to report a number that corresponds to the most desired object, which might differ from the object they had actually rolled. The original survey procedure of the dice-roll task can be seen in the following short video: https://osf.io/v49tq/.

Since students could report a more and a less valuable object from their subjective preference list relative to the number they rolled, there were multiple ways of misreporting the rolled number. Consequently, there were many different ways to empirically define students' misreports:

i. Misreport = 1 if the rolled object's rank differed from the reported object's rank (regardless of whether students choose a more or a less valuable object); otherwise, misreport = 0. The classification that results from this coding is referred to as overall *cheating*, which was our preregistered *primary outcome*.

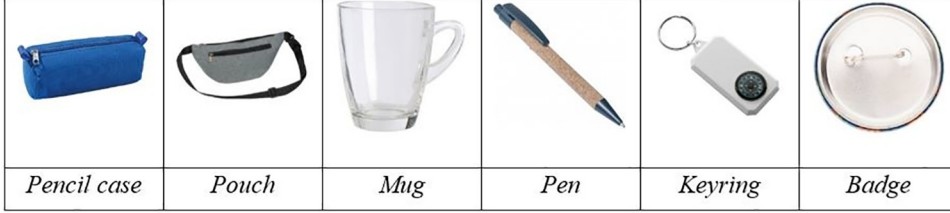

**Fig 1. The objects (gifts) used in the dice-roll exercise as incentives.**

| | Students' preference lists (lower rank means higher preference) | | | | | | Rolled object | Reported object | Rank of rolled | Rank of reported | Rank difference | Misreporting |
| | 1 | 2 | 3 | 4 | 5 | 6 | | | | | | |
| | most preferred <= | | <= <=<= | <= | <= | <= least preferred | | | | | | |
| ID 1 | Pouch | Pencil case | Keyring | Mug | Badge | Pen | Mug | Mug | 4 | 4 | 0 | No misreport |
| ID 2 | Pen | Pencil case | Pouch | Mug | Keyring | Badge | Pencil case | Pouch | 2 | 3 | -1 | Opted for a less valuable object |
| ID 3 | Pencil case | Mug | Pen | Pouch | Keyring | Badge | Badge | Pencil case | 6 | 1 | 5 | Opted for a more valuable object |

**Fig 2. Examples of (mis)reporting the rolled number.**

ii. Misreport = 1 if students indicated a more valuable object; misreport = 0 if students chose the rolled object; the variable has missing values if students indicated a less valuable object. The classification that results from this coding is referred to as *cheating* to obtain a more valuable object, which is our preregistered *secondary outcome*.

iii. Misreport = 1 if students indicated a less valuable object; misreport = 0 if students chose the rolled object; the variable has missing values if students indicated a more valuable object. Since choosing a subjectively less valuable object is irrational, the variable resulting from this classification most likely measures inattention rather than cheating.

Fig 2 shows examples of (mis)reporting the rolled numbers.

In our data, the rolled number was misreported by 12.6% of students (N = 144). Most of the misreporters (N = 97) indicated a more valuable object, a sign of purposeful cheating. However, a smaller share of students (N = 47) indicated a less valuable object, possibly due to inattention.

## 2.4 The treatment exposure

The treatment exposure consisted of a 20-minute-long questionnaire to deplete the students. The questionnaire depleted students' resources in various ways. First, the math test required cognitive effort [7], which consumed students' self-control in a similar way to a low-stakes assignment (see the S1 Appendix on sample questions used in the math test). Second, the questionnaire required students to concentrate for 20 minutes, challenging their self-control. In sum, the treatment exposure exhausted students similar to the depletion that students experience in school.

## 2.5 Randomization and balance

Based on a value of a randomly generated number, we experimentally manipulated when students would answer the dice-roll task, i.e., at the beginning or at the end of the 20-minute-long questionnaire. Thus, in the control group, students first completed the dice-roll task and then the questionnaire. Therefore, students were not depleted in the control group. In the treatment group, however, we reversed the order. Students first completed the questionnaire and then the dice-roll task. Therefore, students were depleted in the treatment group.

We purposefully designed the size of the treatment groups to be larger (N = 789, 69%) than the size of the control group (N = 354; 31%), because the collected data was also used for a different study, where we investigated short-term changes in students' attitudes [22]. That research question required a stable ordering of questionnaire items.

Randomization was fairly successful (see S3 Table). There was no significant difference between the control group and the treated group in the baseline variables we collected, such as age, number of books at home, GPA (January 2020), teacher-reported disruptive school behavior (February 2020), math test score (April 2020), delay of gratification (April 2020), or

altruism (April 2020). However, there were significantly fewer (8.4 pp) girls in the treated group. Therefore, we control for gender in all of our later estimations.

## 2.6 Hypotheses

We preregistered the hypothesis that (exhausted) students would cheat more in the treated group (in which students received the dice-roll task after the 20-minute-long questionnaire) than in the control group (in which students received the dice-roll task before the 20-minute-long questionnaire).

## 2.7 Statistical analysis

To test our hypothesis, we estimated the following classroom-fixed-effect linear probability model:

$$Cheat_{sc} = \beta_0 + \beta_1 \times T_{sc} + \beta_2 \times X_{sc} + \theta_c + \varepsilon_{sc}$$

Where $Cheat_s$ is the binary measure of cheating concerning student $s$ in class $c$. $T_{sc}$ is a binary variable equal to 1 if student $s$ from classroom $c$ is in the treated group. $\theta_c$ denotes classroom-fixed effects. We include a set of control variables ($X_{sc}$). Coefficient $\beta_1$ is the causal effect of the treatment.

As a robustness check, we re-estimated all models using logistic regression (see S4 Table). Results are qualitatively similar.

## 2.8 IRB approval and consent

The study was reviewed and approved by the IRB office at the Center for Social Sciences, Budapest. We obtained consent at multiple points. First, school principals and teachers provided written consent to participate in the study. Second, parents provided written active consent for the retrieval of administrative records via teachers and for their children's participation in the survey. Students received their reported object at the end of the study. The anonymized data file does not allow the researcher to trace individual students' dishonest behavior. Teachers and schools had no access to students' online survey inputs.

## 3. Results

Fig 3 shows the share of those who misreported the rolled number in the treated and control groups.

In Table 2, we report two specifications of each model. The first specification includes the preregistered control variables of gender and age. The second specification includes further controls measured at the baseline (number of books at home, GPA, teacher-reported disruptive school behavior).

By employing a two-sided t-test, we observe a marginally significant difference of about 4.4 percentage points when considering all misreporting (Columns 1 and 2 in Table 2). The marginally significant coefficient might be due to noise generated by those who opted for a less desired object and thus might be insignificant.

When we focus only on those who opted for a more valuable object (Columns 3 and 4 in Table 2), we find that the size of the treatment effect remains the same but becomes significant at 1%. By contrast, there is no significant difference between the treated and control students' outcomes for students who opted for a less valuable object (Columns 5 and 6 in Table 2). This suggests that students' inattention is not associated with the treatment.

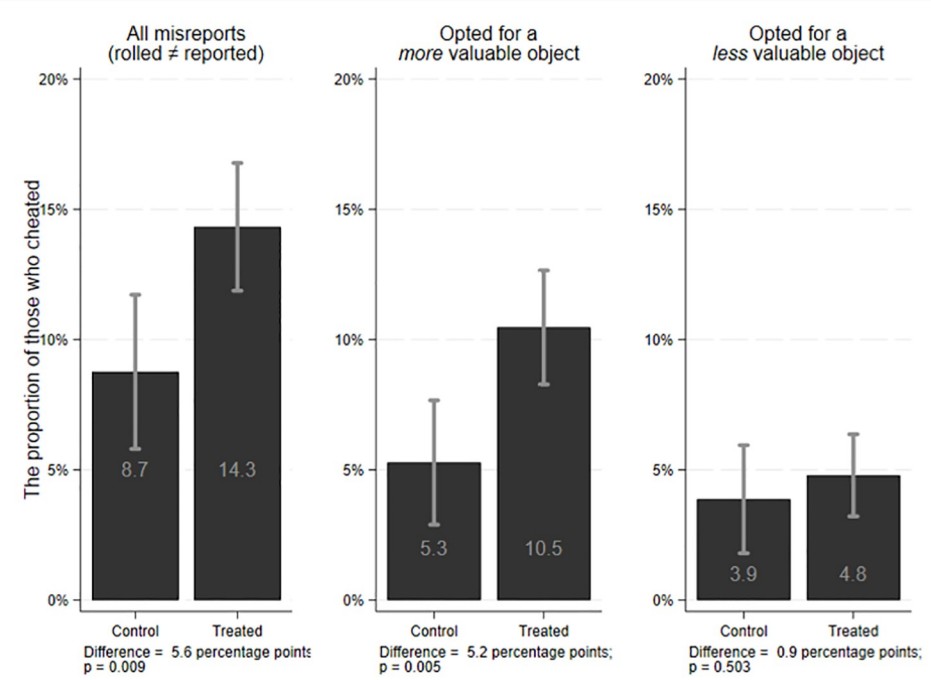

**Fig 3. Distribution of cheating behavior as a function of treatment with 95% confidence interval.**

## 4. Discussion and conclusion

We conducted a large-scale preregistered field experiment to explore the scope conditions of the ego-depletion theory [1]. We found that primary school students whose self-control had been depleted (by doing cognitively demanding exercises) cheated more to achieve the desired outcome in a subsequent task.

**Table 2. Results of regression analyses with linear probability models, unstandardized regression coefficients.**

|  | (1) | (2) | (3) | (4) | (5) | (6) |
|---|---|---|---|---|---|---|
|  | misreported dice roll | | opted for a more valuable object | | opted for a less valuable object | |
|  | Control: preregistered | Control: full | Control: preregistered | Control: full | Control: preregistered | Control: full |
| Treated | 0.044+ (0.024) | 0.045+ (0.024) | 0.044* (0.016) | 0.045** (0.016) | 0.002 (0.019) | 0.002 (0.019) |
| Observations | 1,143 | 1,143 | 1,096 | 1,096 | 1,046 | 1,046 |
| R-squared | 0.144 | 0.153 | 0.158 | 0.165 | 0.130 | 0.141 |
| Cohen's d effect size | 0.132 | 0.135 | 0.155 | 0.160 | 0.010 | 0.009 |
| Mean in the control group | 0.088 | 0.088 | 0.053 | 0.053 | 0.039 | 0.039 |

All models contain constant classroom-fixed effects and the preregistered control variables: gender and age. Standard errors were clustered at the school level.

The list of baseline control variables in *full* specifications is as follows: gender, age, N of books, GPA, teacher-reported disruptive school behavior, math test, delay of gratification (DG), baseline altruism. Missing values in baseline control variables have been replaced with 0, and separate dummy variables control for missing status. Descriptive statistics and the coding of baseline control variables are shown in Table 1.

Two-sided t-test are used. Robust standard errors in parentheses.

** p<0.01

* p<0.05, + p<0.1.

In Model 3, the significance level of the treatment coefficient is 0.0119. Thus, the coefficient is significant after correcting the significance level for multiple testing since 0.0119 < 0.05/3 [= 0.0167].

Our results have substantive, methodological, and practical consequences. Substantively, we explored some specific conditions under which ego depletion occurs. Primary school students' self-control can be depleted with simple assignments, and students with depleted self-control engage in fraudulent behavior.

Methodologically, our experiment informs subsequent research that the ego-depletion effect can be explored beyond the sequential task paradigm and in a real-life situation such as school assignments. Nevertheless, the Cohen's d effect size of the ego-depletion effect we explored (between 0.134 and 0.157) was smaller in size (approximately 25%) than the effect size reported in the first meta-analysis by Hagger [3].

Finally, our findings have practical consequences. Teacher-written assignments often test students at the end of the school day after four or five 45-minute classes. Our results suggest that students might be tempted to cheat during these tests as they are depleted. Teachers should consider scheduling assignments at the beginning of the school day, when students are not exhausted.

Our online experiment is not free from limitations. We had limited scope to control for environmental factors (e.g., internet speed, IT device, home context), potentially resulting in an increase in noise that may have led to less accurate estimates. Nevertheless, our study should reassure the skeptics of ego-depletion theory since it was based on a large-scale and pre-registered experiment. Thus, small sample size and specification search—which are often considered sources of bias in prior empirical works—do not impact our results. The depletion of students' self-control in schools is an issue that requires more future research.

## Supporting information

**S1 Table. Mean differences in students' baseline grades between those who answered/did not answer the online survey.**
(DOCX)

**S2 Table. Differences between students in schools that are/are not in our experiment based on 6$^{th}$-grade students' data in a nationwide administrative dataset.**
(DOCX)

**S3 Table. Balance in the sample: Mean of baseline variables in the control group and treated group relative to the control group.**
(DOCX)

**S4 Table. Results of regression analysis with conditional logit model, logit coefficients.**
(DOCX)

**S1 Appendix. Sample questions used in the math test.**
(DOCX)

## Author Contributions

**Conceptualization:** Tamás Keller, Hubert János Kiss.

**Data curation:** Tamás Keller.

**Formal analysis:** Tamás Keller.

**Funding acquisition:** Tamás Keller, Hubert János Kiss.

**Investigation:** Tamás Keller.

**Methodology:** Tamás Keller.

**Project administration:** Tamás Keller.

**Resources:** Tamás Keller.

**Software:** Tamás Keller.

**Supervision:** Tamás Keller.

**Validation:** Tamás Keller.

**Visualization:** Tamás Keller.

**Writing – original draft:** Tamás Keller, Hubert János Kiss.

**Writing – review & editing:** Tamás Keller, Hubert János Kiss.

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
