## [Decision Letter · Decision Letter 0]

18 Aug 2021

PONE-D-21-23313

Do exhausted primary school students cheat more? A randomized field experiment

PLOS ONE

Dear Dr. Keller,

Thank you for submitting your manuscript to PLOS ONE. After careful consideration, we feel that it has merit but does not fully meet PLOS ONE’s publication criteria as it currently stands. Therefore, we invite you to submit a revised version of the manuscript that addresses the points raised during the review process.

Please find below the reviewer's comments, as well as those of mine.

We look forward to receiving your revised manuscript.

Kind regards,

Valerio Capraro

Academic Editor

PLOS ONE

Journal Requirements:

Additional Editor Comments:

I have now collected one review from one expert on the field, whom I thank for their detailed and thoughtful feedback. I was unable to find a second reviewer. However the review I could collect is very detailed and I am myself familiar with the topic of this manuscript. Therefore, I feel confident in making a decision with only one review. The reviewer thinks that the paper has potential, but suggests a major revision. I agree with the reviewer, therefore I would like to invite you to revise your work for Plos One. Apart from the reviewer's comments, I would like to add two additional comments. I have noticed that you do not review the literature on time pressure and cheating, which I think is very relevant (Gunia et al. 2012; Shalvi et al. 2012; Capraro, 2017; Lohse, Simon & Konrad, 2018; Capraro, Schulz & Rand, 2019). Moreover, there is one review article (Capraro, 2019) and one meta-analysis (Kobis et al. 2020) on the role of System 1/System 2 on cheating and other social behaviours, which I think are relevant too.

I am looking forward for the revision.

Capraro, V. (2017). Does the truth come naturally? Time pressure increases honesty in one-shot deception games. Economics Letters, 158, 54-57.

Capraro, V. (2019). The dual-process approach to human sociality: A review. Available at SSRN 3409146.

Capraro, V., Schulz, J., & Rand, D. G. (2019). Time pressure and honesty in a deception game. Journal of Behavioral and Experimental Economics, 79, 93-99.

Gunia, B. C., Wang, L., Huang, L. I., Wang, J., & Murnighan, J. K. (2012). Contemplation and conversation: Subtle influences on moral decision making. Academy of Management Journal, 55(1), 13-33.

Köbis, N. C., Verschuere, B., Bereby-Meyer, Y., Rand, D., & Shalvi, S. (2019). Intuitive honesty versus dishonesty: Meta-analytic evidence. Perspectives on Psychological Science, 14(5), 778-796.

Lohse, T., Simon, S. A., & Konrad, K. A. (2018). Deception under time pressure: Conscious decision or a problem of awareness?. Journal of Economic Behavior & Organization, 146, 31-42.

Shalvi, S., Eldar, O., & Bereby-Meyer, Y. (2012). Honesty requires time (and lack of justifications). Psychological science, 23(10), 1264-1270.

Reviewers' comments:

Reviewer's Responses to Questions

**Comments to the Author**

1. Is the manuscript technically sound, and do the data support the conclusions?

Reviewer #1: Yes

2. Has the statistical analysis been performed appropriately and rigorously? 

Reviewer #1: Yes

3. Have the authors made all data underlying the findings in their manuscript fully available?

Reviewer #1: Yes

4. Is the manuscript presented in an intelligible fashion and written in standard English?

Reviewer #1: Yes

5. Review Comments to the Author

Reviewer #1: The paper studies the effects of ego depletion on cheating using an online experiment involving Hungarian students aged 9-16. Self-control is depleted using a cognitively demanding task; cheating is measured on a variant of the dice-roll task.

The authors find that there is an ego depletion effect, as those who depleted self-control are more likely to cheat.

There is no consensus in the literature on the ego depletion effect. The authors contribute to the literature in three main directions: 1) they design a task that goes beyond the common sequential task paradigm; 2) they focus on a cheating outcome; 3) they exploit a large-scale experiment involving more than 1,000 students.

I generally like the research question and the experimental setting. The empirical analysis is very simple but neat. I report below some comments, in no particular order, that I hope can help to improve further versions of the paper.

- I learn from Table A4 that the age range of the subjects is 9-16. Since the text repeatedly refers to "primary school students", I expected the subjects to be younger. I think some comments should be made on the Hungarian education system that, as far as I understand, is quite peculiar. Moreover, I believe that information on the age range is more relevant than the school level. For this reason, I would like the text to refer to the age range from time to time. I realized what the age range is only looking at Table A4.

- Three different definitions of cheating are considered. The third one (cheating toward less valuable objects), however, is irrational and probably measures mistakes rather than cheating. I suggest you to completely remove the third definition from the analysis.

- Two sentences in the text raise concerns on the representativeness of the sample. First, the pool of potential subjects perform worse at school and has lower status than the average Hungarian ("On average, students’ test scores and socioeconomic status is lower in our sample than the Hungarian average (see Table A1 in the Appendix)." Second, those who agreed to participate to the experiment perform better at school than those who did not agree to participate ("On average, students who participated in our survey had better grades and better school behavior than their classmates (see Table A2 in the Appendix)." I would like to read comments on how the non-representativeness of the pool of potential subjects and the self-selection into the sample may affect the results.

- You write "In sum, we tempted students to report a higher number than they had rolled to receive a more desirable object". However, if I understand correctly the task, students are tempted to report a lower number (the lower number, the better the prize).

- Is there a reason why you chose to have the treatment group twice as large as the control group?

- Below Table 1 you write that "Descriptive statistics and the coding of baseline control variables is shown in Table A2 in the Appendix". I think you refer to Table A4. However, I do not see there a definition of each control variable. In particular, could you describe the variable "Number of books at home"? I expected this variable to take natural values (0, 1, 2, ...), but I realize it also takes negative ones (see Table A4). Based on the table, it seems you standardized the variable. Why?

- Table 1, columns 1-2. The coefficient is significant only at 10%. Many researchers would tell that a 10% significance is is not a significance, especially if you consider the large sample size you have. From the comparison of all the columns, it seems you cannot get significant results here because of the noise brought by those who cheated toward less valuable objects.

- You write "We modified the standard dice-roll experiment to collect individual data about students’ dishonest behavior while keeping their identities hidden.", but you do not go more in detail. I hope you did not deceive subjects, as deception in experimental economics is (as a matter of fact) banned. I would like to read a comment on this in Section II.

- Why did you use a linear OLS model rather than a non-linear logit/probit model, which is more appropriate when you deal with binary dependent variables?

6. PLOS authors have the option to publish the peer review history of their article (what does this mean?). If published, this will include your full peer review and any attached files.

Reviewer #1: No

---

## [Author Response · Author response to Decision Letter 0]

23 Sep 2021

Reviewer 

Journal Requirements:

1. Please ensure that your manuscript meets PLOS ONE’s style requirements, including those for file naming. The PLOS ONE style templates can be found at 

Thank you for your guidance. We have checked the documents and formatted the manuscript accordingly. Most importantly, we use “Vancouver” style of reference, line numbers, and double-spaced layout.

Data and analytical scripts are publicly available at the projects’ OSF homepage: 

We have included the following section in the manuscript: 

IRB approval and consent 

The study was reviewed and approved by the IRB office at the Center for Social Sciences, Budapest. We obtained consent at multiple points. First, school principals and teachers provided written consent to participate in the study. Second, parents provided written active consent for the retrieval of administrative records via teachers and for their children’s participation in the survey. Students received their reported object at the end of the study. The anonymized data file does not allow the researcher to trace individual students’ dishonest behavior. Teachers and schools had no access to students’ online survey inputs. 

Additional Editor Comments:

I have now collected one review from one expert on the field, whom I thank for their detailed and thoughtful feedback. I was unable to find a second reviewer. However the review I could collect is very detailed and I am myself familiar with the topic of this manuscript. Therefore, I feel confident in making a decision with only one review. The reviewer thinks that the paper has potential, but suggests a major revision. I agree with the reviewer, therefore I would like to invite you to revise your work for Plos One. Apart from the reviewer’s comments, I would like to add two additional comments. I have noticed that you do not review the literature on time pressure and cheating, which I think is very relevant (Gunia et al. 2012; Shalvi et al. 2012; Capraro, 2017; Lohse, Simon & Konrad, 2018; Capraro, Schulz & Rand, 2019). Moreover, there is one review article (Capraro, 2019) and one meta-analysis (Kobis et al. 2020) on the role of System 1/System 2 on cheating and other social behaviours, which I think are relevant too.

I am looking forward for the revision.

Capraro, V. (2017). Does the truth come naturally? Time pressure increases honesty in one-shot deception games. Economics Letters, 158, 54-57.

Capraro, V. (2019). The dual-process approach to human sociality: A review. Available at SSRN 3409146.

Capraro, V., Schulz, J., & Rand, D. G. (2019). Time pressure and honesty in a deception game. Journal of Behavioral and Experimental Economics, 79, 93-99.

Gunia, B. C., Wang, L., Huang, L. I., Wang, J., & Murnighan, J. K. (2012). Contemplation and conversation: Subtle influences on moral decision making. Academy of Management Journal, 55(1), 13-33.

Köbis, N. C., Verschuere, B., Bereby-Meyer, Y., Rand, D., & Shalvi, S. (2019). Intuitive honesty versus dishonesty: Meta-analytic evidence. Perspectives on Psychological Science, 14(5), 778-796.

Lohse, T., Simon, S. A., & Konrad, K. A. (2018). Deception under time pressure: Conscious decision or a problem of awareness?. Journal of Economic Behavior & Organization, 146, 31-42.

Shalvi, S., Eldar, O., & Bereby-Meyer, Y. (2012). Honesty requires time (and lack of justifications). Psychological science, 23(10), 1264-1270.

Thank you very much for the overall positive evaluation of the manuscript. We have taken your advice and cite this relevant body of literature at the end of the introduction. We say that: 

“The results imply that dishonesty but not honesty is intuitive since resisting the temptation of dishonesty needs resources that might be consumed in tasks requiring self-control. Nevertheless, our paper has a narrow focus on the ego-depletion effect. We acknowledge but leave apart from our focus the more general scientific debates on intuitive honesty/dishonesty (Köbis et al. 2019) and the related tests on time pressure (Capraro, Schulz, and Rand 2019; Shalvi, Eldar, and Bereby-Meyer 2012) or cognitive load (Welsh and Ordóñez 2014) that bring (conflicting) experimental evidence on this debate. “

5. Review Comments to the Author

Reviewer #1: The paper studies the effects of ego depletion on cheating using an online experiment involving Hungarian students aged 9-16. Self-control is depleted using a cognitively demanding task; cheating is measured on a variant of the dice-roll task.

The authors find that there is an ego depletion effect, as those who depleted self-control are more likely to cheat.

There is no consensus in the literature on the ego depletion effect. The authors contribute to the literature in three main directions: 1) they design a task that goes beyond the common sequential task paradigm; 2) they focus on a cheating outcome; 3) they exploit a large-scale experiment involving more than 1,000 students.

I generally like the research question and the experimental setting. The empirical analysis is very simple but neat. I report below some comments, in no particular order, that I hope can help to improve further versions of the paper.

- I learn from Table A4 that the age range of the subjects is 9-16. Since the text repeatedly refers to “primary school students”, I expected the subjects to be younger. I think some comments should be made on the Hungarian education system that, as far as I understand, is quite peculiar. Moreover, I believe that information on the age range is more relevant than the school level. For this reason, I would like the text to refer to the age range from time to time. I realized what the age range is only looking at Table A4.

Thank you very much for this comment which motivated us to move Table A4 to the main body of the manuscript (the updated Table A4 is referred to as Table 1 in the recent manuscript). We briefly refer to the Hungarian educational system and provide information on the age range of students. We say in the updated manuscript that 

“Primary education in Hungary is compulsory and encompasses the primary and lower-secondary ISCED 1 and ISCED 2 levels, comparable to elementary and middle school in the United States.” 

Furthermore: 

“The age of students ranged from 10 to 16 years (mean = 12.82, standard deviation = 1.43). The high maximum age is due to students who had to repeat classes.”

- Three different definitions of cheating are considered. The third one (cheating toward less valuable objects), however, is irrational and probably measures mistakes rather than cheating. I suggest you to completely remove the third definition from the analysis.

Motivated by your valuable insight, we do not call cheating when students choose a less valuable object, but we refer to this as inattention. Nevertheless, since this outcome was preregistered, we show the result. However, we have toned down the language used to discuss the results on this outcome. 

- Two sentences in the text raise concerns on the representativeness of the sample. First, the pool of potential subjects perform worse at school and has lower status than the average Hungarian (“On average, students’ test scores and socioeconomic status is lower in our sample than the Hungarian average (see Table A1 in the Appendix).” Second, those who agreed to participate to the experiment perform better at school than those who did not agree to participate (“On average, students who participated in our survey had better grades and better school behavior than their classmates (see Table A2 in the Appendix).” I would like to read comments on how the non-representativeness of the pool of potential subjects and the self-selection into the sample may affect the results.

We have elaborated on the issues of self-selection and non-representativeness in the updated manuscript. We write that:

“The self-selected students who participated in our survey may have differed from their nonrespondent classmates in motivation—that is, in terms of the motivation that may have prompted students to engage with the purpose of the study and self-select into the sample. Motivation might render immunity to ego depletion, but only if the experienced depletion is mild—suggesting that there is a limit to the influence of motivation on ego-depletion (Vohs, Baumeister, and Schmeichel 2012). Therefore, our reliance on a sample containing self-selected motivated primary school students might have introduced limited bias in estimating the ego-depletion effect. 

The demographic composition of the analytical sample leads to an over-representation of students with weaker academic achievement and from poorer social backgrounds. Thus, students in our sample might have lower initial self-control, since low academic achievement translates into low self-control (Duckworth et al. 2019). Nevertheless, the ego-depletion effect is weaker among low self-control students who have experience in resisting acute temptation and have gained experience in how to resist temptation (in contrast to high self-control students who might lack these experiences) (Imhoff, Schmidt, and Gerstenberg 2014). Therefore, our reliance on a sample containing self-selected motivated primary school students might have introduced limited bias in estimating the ego-depletion effect.” 

- You write “In sum, we tempted students to report a higher number than they had rolled to receive a more desirable object”. However, if I understand correctly the task, students are tempted to report a lower number (the lower number, the better the prize).

Thank you for your careful reading! We have corrected the inconsistency and elaborated more on the section describing the measurement of cheating. 

- Is there a reason why you chose to have the treatment group twice as large as the control group?

We purposefully designed the size of the treatment groups to be larger (N=789, 69%) than the size of the control group (N=354; 31%), because the data collected was also used for a different study, where we investigated short-term changes in students’ attitudes(Kiss and Keller 2021). That research question required a stable ordering of questionnaire items. 

”

- Below Table 1 you write that “Descriptive statistics and the coding of baseline control variables is shown in Table A2 in the Appendix”. I think you refer to Table A4. However, I do not see there a definition of each control variable. In particular, could you describe the variable “Number of books at home”? I expected this variable to take natural values (0, 1, 2, ...), but I realize it also takes negative ones (see Table A4). Based on the table, it seems you standardized the variable. Why?

We moved Table A4 to the main body of the manuscript (the updated Table A4 is referred to as Table 1 in the recent manuscript) and we provide a detailed description of the deployed variables. 

- Table 1, columns 1-2. The coefficient is significant only at 10%. Many researchers would tell that a 10% significance is is not a significance, especially if you consider the large sample size you have. From the comparison of all the columns, it seems you cannot get significant results here because of the noise brought by those who cheated toward less valuable objects.

Motivated by your comment, we have changed the langue that discusses the main results. We say that “The marginally significant coefficient might be due to noise generated by those who opted for a less desired object and thus might be insignificant.”

- You write “We modified the standard dice-roll experiment to collect individual data about students’ dishonest behavior while keeping their identities hidden.”, but you do not go more in detail. I hope you did not deceive subjects, as deception in experimental economics is (as a matter of fact) banned. I would like to read a comment on this in Section II.

We added a section that describes the parental consent we obtained and provides information about the attained IRB consent. 

- Why did you use a linear OLS model rather than a non-linear logit/probit model, which is more appropriate when you deal with binary dependent variables?

As a robustness check, we re-estimated all models using logistic regression (see S4 Table). Results are qualitatively similar.

---

## [Decision Letter · Decision Letter 1]

6 Oct 2021

PONE-D-21-23313R1Do exhausted primary school students cheat more? A randomized field experimentPLOS ONE

Dear Dr. Keller,

Thank you for submitting your manuscript to PLOS ONE. After careful consideration, we feel that it has merit but does not fully meet PLOS ONE’s publication criteria as it currently stands. Therefore, we invite you to submit a revised version of the manuscript that addresses the points raised during the review process.

We look forward to receiving your revised manuscript.

Kind regards,

Valerio Capraro

Academic Editor

PLOS ONE

Journal Requirements:

Additional Editor Comments (if provided):

The reviewer suggests one final comment before publication. Please address this comment at your earliest convenience. I am looking forward to receiving the final version.

Reviewers' comments:

Reviewer's Responses to Questions

**Comments to the Author**

1. If the authors have adequately addressed your comments raised in a previous round of review and you feel that this manuscript is now acceptable for publication, you may indicate that here to bypass the “Comments to the Author” section, enter your conflict of interest statement in the “Confidential to Editor” section, and submit your "Accept" recommendation.

Reviewer #1: All comments have been addressed

2. Is the manuscript technically sound, and do the data support the conclusions?

Reviewer #1: Yes

3. Has the statistical analysis been performed appropriately and rigorously? 

Reviewer #1: Yes

4. Have the authors made all data underlying the findings in their manuscript fully available?

Reviewer #1: Yes

5. Is the manuscript presented in an intelligible fashion and written in standard English?

Reviewer #1: (No Response)

6. Review Comments to the Author

Reviewer #1: The authors paid attention to my previous comments and did a good job in revising the paper.

Please notice that on line 311 you refer to Table 1, whereas the correct reference is now Table 2.

I have no further comments.

7. PLOS authors have the option to publish the peer review history of their article (what does this mean?). If published, this will include your full peer review and any attached files.

Reviewer #1: No

---

## [Author Response · Author response to Decision Letter 1]

2 Nov 2021

Journal Requirements:

.

We do not cite Ariely’s retracted paper (Signing at the beginning makes ethics salient and decreases dishonest self-reports in comparison to signing at the end), published in PNAS and retracted in September 2021. The retracted paper concerns cheating from the perspective of self-concept maintenance theory and not from ego-depletion theory. Therefore, we have not cited this research in any of the earlier versions of this manuscript. 

Reviewer’s comment:

The authors paid attention to my previous comments and did a good job in revising the paper.

Please notice that on line 311 you refer to Table 1, whereas the correct reference is now Table 2.

Thank you very much for your careful reading. We have corrected our mistake in line 311.

---

## [Editor Report · Decision Letter 2]

4 Nov 2021

Do exhausted primary school students cheat more? A randomized field experiment

PONE-D-21-23313R2

Dear Dr. Keller,

We’re pleased to inform you that your manuscript has been judged scientifically suitable for publication and will be formally accepted for publication once it meets all outstanding technical requirements.

Kind regards,

Valerio Capraro

Academic Editor

PLOS ONE
---

## [Editor Report · Acceptance letter]

8 Nov 2021

PONE-D-21-23313R2 

Do exhausted primary school students cheat more? A randomized field experiment 

Dear Dr. Keller:

I'm pleased to inform you that your manuscript has been deemed suitable for publication in PLOS ONE. Congratulations! Your manuscript is now with our production department. 

Kind regards, 

on behalf of

Dr. Valerio Capraro 

Academic Editor

PLOS ONE